# Endoscopic Predictors of Neoplastic Lesions in Inflammatory Bowel Diseases Patients Undergoing Chromoendoscopy

**DOI:** 10.3390/cancers14184426

**Published:** 2022-09-12

**Authors:** Elisabetta Lolli, Elena De Cristofaro, Irene Marafini, Edoardo Troncone, Benedetto Neri, Francesca Zorzi, Livia Biancone, Emma Calabrese, Giovanni Monteleone

**Affiliations:** 1Azienda Ospedaliera Policlinico Tor Vergata, 00133 Rome, Italy; 2Department of Systems Medicine, University of Rome “Tor Vergata”, 00133 Rome, Italy

**Keywords:** Crohn’s disease, ulcerative colitis, dysplasia, Kudo pit pattern, colitis-associated colorectal cancer

## Abstract

**Simple Summary:**

Patients with long-standing and extensive/left-sided colonic inflammatory bowel diseases (IBD) have enhanced risk of developing colorectal cancer with respect to the general population. Dye-based chromoendoscopy (DCE) is now considered as the best surveillance strategy to prevent colon cancer in such patients, even though the endoscopic features of the DCE-evidenced lesions that predict neoplasia are not fully clarified. This study was aimed at identifying predictive factors of dysplastic/neoplastic lesions in IBD patients undergoing DCE. Our study shows that polypoid lesions with specific morphologic features and size greater than 7 mm are frequently dysplastic/neoplastic and, therefore, must be removed.

**Abstract:**

Dye-based chromoendoscopy (DCE) with targeted biopsies is recommended for surveillance of patients with long-standing inflammatory bowel diseases (IBD), but endoscopic features that predict dysplasia are not fully clarified. We here aimed at identifying predictive factors of dysplastic/neoplastic lesions in IBD patients undergoing DCE. Two-hundred-and-nineteen patients were consecutively and prospectively enrolled from October 2019 to March 2022. One-hundred-and-forty-five out of 219 patients underwent DCE, and 148 lesions were detected in 79/145 (54%) patients. Thirty-nine lesions (26%) were dysplastic and one of them contained a CRC. Among these lesions, 7 (17.9%) had Kudo pit pattern I-II and 32 (82.1%) had a neoplastic pit pattern (Kudo III-IV). Multivariate analysis showed that neoplastic lesions Kudo III-IV (OR: 5.8, 95% CI: 2.3–14.6; *p* = 0.0002), lesion’s size (OR 1.16, 95% CI: 1.06–1.26; *p* = 0.0009), and polypoid lesions according to Paris Classification (OR 7.4, 95% CI: 2.7–20.2; *p* = 0.0001) were independent predictors of dysplasia. A cut-off of lesion’s size > 7 mm was identified as the best predictor of dysplasia. Among such features, Kudo pit pattern III-IV had the highest sensitivity and specificity to predict dysplasia (79% and 80%, respectively). Lesions with all three endoscopic features had a sensitivity of 90% and specificity of 100% to predict dysplasia. In contrast, non-polypoid lesions were inversely associated with dysplasia (OR 0.13, 95% CI: 0.05–0.36; *p* = 0.0001). These findings indicate that, in IBD patients, DCE-evidenced polypoid lesions with Kudo pit pattern III-IV and size > 7 mm are frequently dysplastic.

## 1. Introduction

Patients with ulcerative colitis (UC) and patients with Crohn’s disease (CD), the major inflammatory bowel diseases (IBD) in humans, are at higher risk of developing colorectal cancer (CRC) relative to the general population [1,2]. In most cases, the precursor lesion of CRC is dysplasia, and many endoscopy and gastroenterology societies recommend performing follow-up by colonoscopy in patients with extensive or left-sided colonic disease to detect dysplasia. The screening colonoscopy program starts at 8–10 years following IBD diagnosis and traditionally the method consisted of sampling of multiple random biopsies throughout the colon to screen for flat undetectable neoplastic lesions [3,4]. This procedure was time consuming, expensive, and often poorly adopted in clinical practice. More recently, the introduction of advanced image-enhanced endoscopy techniques, such as dye-based chromoendoscopy (DCE) with targeted biopsies, has markedly improved the resolution of images compared with conventional approaches [5,6]. Randomized trials demonstrated that DCE is superior to white-light endoscopy with random biopsy sampling for the detection of dysplasia in IBD patients, even though a large, retrospective study showed that DCE was not superior to white-light colonoscopy with targeted or random biopsies in detecting dysplasia in clinical practice [7]. These discrepancies could rely on the fact that with improvement in resolution of images in endoscopy and expertise in optical diagnosis, the advantage of DCE may become less apparent, at least for expert operators. However, an expert panel recently released the SCENIC guidelines, which suggested the routine use of DBC as an adjunct to high-definition colonoscopy [8]. More than 80% of the experts were in agreement with this recommendation, despite the low-grade evidence, while the panelists were unable to reach an agreement about the necessity of random biopsies, with only 60% of the panel members suggesting random biopsies were not necessary when high definition DCE was performed. Nonetheless, several issues remain to be solved. For instance, it remains unclear whether every operator doing surveillance in IBD patients should adopt DCE. Similarly, further work is needed to ascertain the value of virtual chromoendoscopy and untargeted biopsies in the clinical practice. Moreover, conflicting results about endoscopic features that help discriminate between dysplastic and non-dysplastic lesions were reported across various studies, probably reflecting differences in the defined end-points, sample size, study population, presence of active colonic inflammation, and endoscopes used for DCE.

The aim of this prospective study was to further assess endoscopic factors that predict dysplastic/neoplastic lesions in IBD patients undergoing DCE in a real-life scenario.

## 2. Materials and Methods

### 2.1. Participants and Study Design

Consecutive outpatients with a long-standing colonic IBD or patients with primary sclerosing cholangitis undergoing surveillance colonoscopy were prospectively recruited at the Tor Vergata university hospital (Rome, Italy). All the patients were under regular follow-up in our IBD tertiary center and had a clinical indication to perform screening/surveillance colonoscopy [8,9]. Patients were 18 years or older; had a diagnosis of extensive or left-sided UC, colonic CD or unclassified IBD (C-UNC), involving at least one third of colonic mucosa; and a disease duration of at least 8 years since onset of symptoms or any disease duration for patients with concomitant primary sclerosing cholangitis. All patients were able to provide informed consent. Endoscopic remission or mild endoscopic disease activity were defined with a Mayo Endoscopic Score ≤ 1 for UC or SES-CD ≤ 6 for CD [10,11].

The following were considered as exclusion criteria: proctitis or mono-segmentary colonic CD/unclassified IBD, an active endoscopic disease (defined as Mayo endoscopic Score ≥ 2 or SES-CD ≥ 7, with exception of moderate or severe disease activity in a single colonic segment), inability to take bowel preparation, pregnancy, coagulopathy, known allergy to dye, previous colorectal resection, poor bowel preparation defined as a Boston Bowel Preparation Score (BBPS) < 2 in any segment, and incomplete colonoscopy (defined as unsuccessful cecal intubation).

At the time of screening/surveillance endoscopy, the following demographic and clinical data were collected: gender, age, type of IBD, disease duration, extent and phenotype according to Montreal Classification [12], risk factors for IBD (e.g., smoking habit, previous appendectomy, family history of IBD), risk factors for colorectal cancer (e.g., history of previous dysplasia, familiar history of CRC, concomitant primary sclerosing cholangitis), and previous and concomitant use of thiopurines or biologics. The study was approved by the local Ethics Committee (N. 106.22).

### 2.2. Endoscopic and Histopathological Assessment

Colonoscopy was requested by the referring gastroenterologist according to clinical practice and current guidelines. All patients underwent bowel preparation with split-dose polietilenglicole. All colonoscopies were scheduled as outpatient procedures and performed using high-definition colonoscopes (Olympus 185 series) by trained endoscopists. Bowel preparation was graded according to BBPS [13]. DCE was performed using Methylene Blue 0.04% or Indigo Carmine 0.03% injected circumferentially during the withdrawal phase with water-jet pump, and each segment of colonic mucosa was meticulously inspected. The withdrawal time, excluding intervention time, was recorded. Two biopsies for each colonic segment were routinely sampled for the assessment of histological activity. The target dye sprayed was applied on the mucosal areas that appeared different from the colonic mucosa in terms of color and pattern, and the suspected dysplastic lesions were endoscopically removed.

According to the Paris classification, the lesions’ morphology was defined as follows: 0-I, elevated or polypoid lesions (0-Ip Polypoid/pedunculated; 0-Is Polypoid/sessile); 0-II flat or superficial lesions (0-IIa flat and elevated; 0-IIb completely flat; 0-IIc superficially depressed); laterally spreading tumor (LST), flat lesions > 10 mm [14]. According to the Kudo classification, the pit pattern of the lesions was defined as follows: Kudo I, normal mucosa; Kudo II, hyperplasia; Kudo III (IIIs, IIIL) and IV, intramucosal lesions; Kudo V (Vl, Vn), lesions with mucosal and submucosal deep invasion characterized by irregular pattern [15].

In order to reduce the variability, the lesions were divided in polypoid and non-polypoid based on their morphology, and neoplastic (Kudo III-IV-V) and non-neoplastic (Kudo I-II) based on their pit pattern. Each sample was stored in a separate bottle with formalin 10% and the presence of dysplasia was assessed in all the biopsies.

### 2.3. Statistical Analysis

Qualitative data were expressed as number and proportion (%) and quantitative data were expressed as average ± standard deviation or median (range). Patient’s characteristics were compared by using χ square test or exact Fisher test for categoric variables and Mann–Whitney test or Student’s *t*-test for con continuous variables. A logistic regression was performed, and the parameters with *p* < 0.05 in the univariate analysis were used to perform a multivariate logistic regression analysis to determine their influence on risk of dysplasia. The results of the logistic regression analysis were expressed using odds ratios (ORs) and 95% confidence intervals (CIs) with the *p* values. Receiver operating characteristic (ROC) curve was plotted to identify lesion’s size cut-off. *p* value < 0.05 was considered statistically significant. Statistical analysis was performed by using GraphPad Prism version 9.0.

## 3. Results

### 3.1. Study Population

From October 2019 to March 2022, 219 consecutive outpatients with either a long-standing IBD or concomitant primary sclerosing cholangitis underwent endoscopic assessment. Seventy-four out of 219 patients (33.8%) were excluded for the following reasons: disease activity (*n* = 28; 37.9%), poor bowel preparation according to BBPS (*n* = 43, 58%), colonic stricture (*n* = 1; 1.4%), and temporary unavailability of high-definition scopes (*n* = 2; 2.7%) (Figure 1). Upon application of the exclusion criteria, 145 out of 219 (66.2%) patients underwent DCE and were included in the analysis.

Demographic and clinical characteristics of the study population are shown in Table 1. The median age was 53 years (range 20–80 years), and 82 patients were male (56.6%). The median disease duration was 20 years (range 2–62 years), with median age at disease onset of 30 years (range 3–64 years). One hundred and twenty-five patients had UC (86.2%): among them, 49 patients (39.2%) had left-sided colitis and 76 patients (60.8%) an extensive colitis. Sixteen patients had colonic CD (11.1%) and four patients (2.7%) a C-UNC. Nearly half of the patients (76/145, 52.4%) had a low risk for CRC and 24 patients (16.6%) had a familiar history of CRC.

### 3.2. Endoscopic Findings and Histologic Assessment

Endoscopic characteristics of the 145 patients undergoing DCE are shown in Table 2. Seventy-two UC patients (57.6%) were in endoscopic remission (Mayo score 0), 20 UC patients (16%) had a mild endoscopic activity (Mayo score 1), and 22 (17.6%) and 11 patients (8.8%) had a moderate and severe activity (Mayo score 2–3) limited to rectum, respectively. The median SES-CD in the 16 CD patients was 1.5 (range 0–11). The four patients with C-UNC were in endoscopic remission.

Median BBPS was 9 (range 6–9). In 125 out of 145 (86%) patients, the bowel preparation was excellent (BBPS 8–9), and in 20 patients (14%) was classified as good (BBPS 6–7). Median withdrawal time was 22 min (range 8–70). The most used dye was methylene blue [131/145 patients (90.3%)].

One hundred and fifty-two visible lesions were found in 79 patients (54.5%). One hundred and thirteen out of the 152 visible lesions (74.3%) were endoscopically removed, but four of them (3.5%) where not retrieved for histology, so they were excluded from the analysis (Figure 2). In 39 out of 152 lesions (25.7%), targeted biopsy samples were taken. The lesions were classified according to Paris classification as: sessile polypoid 0-Is [74 (50%)], sub peduncolated polypoid 0-Isp [3 (2%)], slightly elevated non polypoid 0-IIa [52 (35.2%)], flat non polypoid 0-IIb [16 (10.8%)], Laterally Spreading Tumour (LST) [3 (2%)]. According to Kudo classification, 95 lesions were classified as non-neoplastic Kudo I-II (64.2%) and 53 as neoplastic Kudo III-IV (35.8%). Two representative endoscopic pictures of neoplastic and non-neoplastic lesions are shown in Figure 3. Overall, a diagnosis of dysplasia was made in 39 out of 148 lesions (26.4%), while the remaining 109 lesions (73.6%) were hyperplastic. Thirty-two out of 39 (82%) dysplastic lesions were found in patients with excellent bowel preparation and 7 lesions (18%) in those with good bowel preparation.

### 3.3. Comparison between Patients with Dysplastic and Non-Dysplastic Lesions

Overall, dysplastic lesions were found in 28 out of 145 patients (19.3%) undergoing DCE, while non-dysplastic lesions were seen in 117 patients (80.7%). Table 3 shows the differences between patients in the dysplastic group and those in the non-dysplastic group. The median age was significantly greater in the group of patients with dysplastic lesions (62.5 years, range 35–78) as compared to the group of patients with non-dysplastic lesions (53 years, range 20–80) (*p* = 0.0025). Similarly, at disease onset, patients with dysplastic lesions were older (median age 38 years, range 5–62) than patients with non-dysplastic lesions (27.5 years, range 3–64) (*p* = 0.0012). When analysis was restricted to patients with defined risk for CRC, the percentage of patients at high risk with dysplastic lesions (13/28, 46.4%) was significantly greater than that of patients at high risk without dysplastic lesions (17/117, 14.5%; *p* = 0.0005). Moreover, dysplastic lesions were less common in patients with a familiar history of IBD (Table 3). No further significant differences were seen between the two groups.

### 3.4. Predictors of Dysplasia

In the dysplasia group, the Kudo pit pattern I-II and Kudo III-IV lesions were identified in 7 and 32 lesions, respectively, while, in the group without dysplasia the Kudo pit pattern I-II and III-IV were identified in 87 and 22 lesions, respectively (*p* < 0.0001). The median value of lesion size was 7 mm in dysplastic lesions and 5 mm in non-dysplastic group (*p* < 0.0001). In the dysplastic group, 28 out of 39 lesions (72%) were classified as polypoid (25 0-Is and 3 0-Isp) and 11 (28%) as non-polypoid (8 IIa, 1 IIb and 2 LST), while among the 109 non-dysplastic lesions, 49 (45%) had polypoid (0-Is) and 60 (55%) a non-polypoid morphology, respectively (*p* = 0.0049). Sixteen dysplastic lesions were located in the right colon, 15 in the transverse colon, and 8 in the left colon including rectum. In the non-dysplastic group, 38 were located in the right colon, 30 in the transverse colon, and 41 in the left colon including rectum. There was no significant difference regarding the colonic localization of the lesions between the two groups (*p* = 0.14)

Univariate analysis showed that neoplastic lesions Kudo III-IV (OR: 6.08, 95% CI 2.77–13.4; *p* < 0.0001), lesion’s size (OR: 1.13, 95% CI 1.14–1.21; *p* = 0.0002), and polypoid lesions (0-Is, 0-Isp, 0-Ip) (OR: 2.93, 95% CI 1.39–6.21; *p* = 0.0037) were statistically associated with dysplastic lesions. In contrast, there was an inverse correlation between dysplasia and non-polypoid lesions (0-IIa, 0-IIb, LST) (OR: 0.34, 95% CI 0.16–0.72; *p* = 0.0047) and rectal localization of the lesions (OR: 0.04, 95% CI 0.01–0.92; *p* = 0.04). Multivariate analysis showed that neoplastic lesions Kudo III-IV (OR: 5.8, 95% CI: 2.3–14.6; *p* = 0.0002), lesion’s size (OR 1.16, 95% CI: 1.06–1.26; *p* = 0.0003), and polypoid lesions (OR 7.4, 95% CI: 2.7–20.2; *p* = 0.0001) were independent predictors of dysplasia, while non-polypoid lesions were inversely associated with dysplasia (OR 0.13, 95% CI: 0.05–0.36; *p* = 0.0001) (Table 4).

To establish which cut-off of lesion size predicted the risk of dysplasia, an ROC curve was plotted. The area under the curve, sensitivity, and specificity were 0.71, 50%, and 85%, respectively. The cut-off value for lesion size was >7 mm (Figure 4). The sensitivity, specificity, the positive predictive value (PPV), and negative predictive value (NPV) of the risk factors to predict dysplasia are shown in Table 5. When combined together, polypoid lesions with Kudo III-IV and size > 7 mm had sensitivity, specificity, PPV, and NPV to predict dysplasia of 90%, 100%, 100%, and 96% respectively.

## 4. Discussion

Although the debate about the best procedure to detect dysplasia in IBD patients is still ongoing, the recent SCENIC guidelines strongly recommend the use of DCE in the performance of surveillance of such patients [8]. However, the difficulty of having high-definition scopes and specific training programmes are still major obstacles to adopt such a strategy in many IBD centers. The present, monocentric prospective study was undertaken to identify endoscopic predictors of dysplastic lesions in outpatients with longstanding IBD or concomitant primary sclerosing cholangitis in a real-life scenario.

The endoscopic lesions with nodular or villous pattern, depressed/ulcerated morphology, dark or uneven redness color, and irregular vascular pattern are most likely dysplastic. Nevertheless, IBD-related inflammation can modify the mucosa appearance, thereby making difficult the detection and characterization of the colonic lesions. As reported by Iacucci and colleagues, DCE has to be performed in high-quality conditions in order to minimize the difficulty to interpret lesions and consequent sampling [14]. In our study, DCE was performed with high-definition scopes, by selecting patients with adequate bowel preparation and in endoscopic remission, in order to minimize difficulties during the evaluation of the colonic lesions in terms of pattern and morphology.

Dysplasia was diagnosed in nearly one fourth of the lesions, which developed in 28 out of the 145 patients, in line with results of previous studies [15,16,17,18]. Nearly one fifth of the patients presented a high- or medium-risk feature for dysplasia, and dysplastic lesions were more common in such patients. The only CRC diagnosed during the study developed in a UC patient with familiar history of CRC and a previously diagnosed dysplasia. This supports the notion that surveillance with DCE in high-risk patients is highly proficient in a real-life setting. DCE is a useful procedure to assess pit patterns, opening shapes of tumor crypts. The classification proposed by Kudo and colleagues includes eight pit pattern types and predicts diagnosis of CRC and tumor depths [19]. According to this classification, pit patterns III-V are considered as neoplastic lesions in non-IBD patients. More recent studies supported the validity of Kudo pit pattern classification to differentiate neoplastic lesions from non-neoplastic lesions in IBD. In a recent, multicentre cohort study including 350 patients and aimed at assessing the effectiveness of DCE for neoplasia detection and characterisation, Carballal and colleagues showed that DCE presents a high diagnostic yield for neoplasia detection. The authors showed also that location of the lesions in the proximal colon, protruding morphology (Paris 0—Ip and 0—Is), loss of innominate lines, and neoplastic pit pattern (IIIs, IIIL, IV, and V) were endoscopic characteristics predictive of dysplasia [20]. Along the same line are the results of Iacucci and colleagues, who confirmed that the Kudo pit patterns IIO and III-V were important predictive features of dysplasia in long-standing IBD patients [18]. Bisschops et al. assessed the accuracy levels of agreement among experts of the Kudo pit pattern in UC with no magnified Narrow Binding Imaging (NBI). It was shown that the assessment of pit pattern I or II with non-magnified high-definition DCE or NBI has a high negative predictive value to rule out neoplasia [21]. However, a somehow different scenario emerges from other studies showing that Kudo pit pattern types IIIL and IV can be observed not only in the neoplastic lesions but also in the surrounding flat mucosa, thus resulting in the low specificity of pit patterns type III-V to diagnose neoplasia [22]. In our study, the different endoscopic parameters were combined in order to increase the dysplasia prediction in IBD patients undergoing surveillance DCE.

Our data indicate that the dysplastic lesions had more frequently a Kudo pit pattern III-IV than a Kudo pit pattern I-II, while the majority of non-dysplastic lesions had a Kudo pit pattern I-II. However, the demonstration that 18% of the dysplastic lesions had a Kudo pit pattern I-II indicates that Kudo pit pattern classification alone is not sufficient to individuate all the dysplastic lesions developing in IBD patients, thereby suggesting the necessity of further variables/features to differentiate between dysplastic lesions and non-dysplastic lesions. Indeed, our multivariate analysis showed that in addition to the Kudo pit pattern III-IV, the size > 7 mm and polypoid morphology of the lesions were independent predictors of dysplasia. Among these variables, Kudo pit pattern III-IV showed the highest sensitivity and specificity to detect dysplasia. Moreover, when these three features were combined together, the sensitivity and specificity increased further, reaching values of 90% and 100% respectively. Our study has some limitations. First, it was a single center study with a relatively small sample size of patients undergoing DCE. In particular, the small number of patients with colonic CD included in the study (*n* = 16) does not allow us to ascertain whether the predictive factors of dysplasia can be generalised to both IBD. Second, no distinction between different grades of dysplasia was made and, therefore, we cannot exclude the possibility that this could have partly influenced some results. Third, the study design did not allow us to answer whether sampling random biopsy samples might increase the detection rate of dysplasia, as recent studies indicate that random biopsies should be taken in association with DCE in IBD patients with personal history of neoplasia, concomitant primary sclerosing cholangitis, or a tubular colon [15]. Finally, we would like to point out that the identified endoscopic factors had already been associated with increased risk of dysplasia [8], even though we here show that combining more variables enhances the sensitivity and specificity to predict dysplastic lesions.

However, our study has some strengths. It was performed in a real-life scenario and included prospectively and consecutively IBD patients referred for dysplasia screening. Moreover, we included a well-characterized cohort of patients with optimal bowel preparation and endoscopic remission except for some cases in which lesions were limited to the rectum. This, together with the use of high-definition endoscopes, allowed us to identify all the lesions and perform targeted biopsies. This data may help adapt surveillance intervals and therapeutic strategies based on endoscopic characteristics. Indeed, the presence of dysplastic lesions requires their complete removal and for patients with such lesions the intervals of endoscopic surveillance have to be closer.

## 5. Conclusions

Our findings indicate that, in IBD patients, DCE-evidenced polypoid lesions with Kudo pit pattern III-IV and size > 7 mm are frequently dysplastic and need to be removed en bloc. The possibility of combining different endoscopic parameters could aid to identify dysplastic lesions and to improve the strategies to prevent IBD-associated colon cancer.

## Figures and Tables

**Figure 1 cancers-14-04426-f001:**
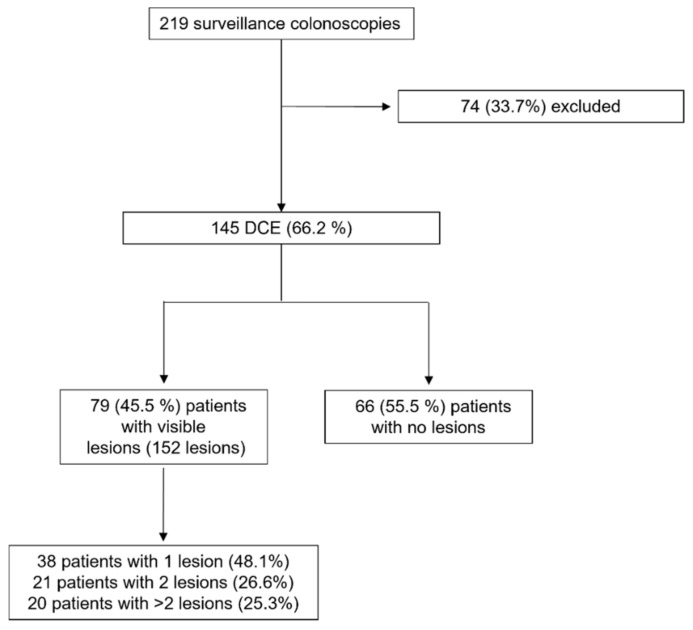
Flow-chart of patients with long-standing colonic IBD or concomitant primary sclerosing cholangitis undergoing surveillance colonoscopy.

**Figure 2 cancers-14-04426-f002:**
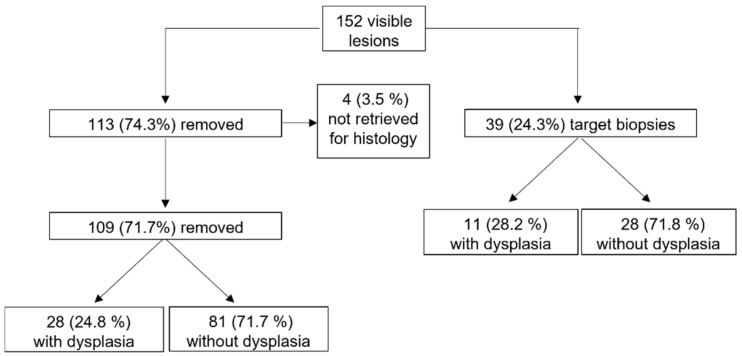
Management and histological assessment of the lesions found during DCE.

**Figure 3 cancers-14-04426-f003:**
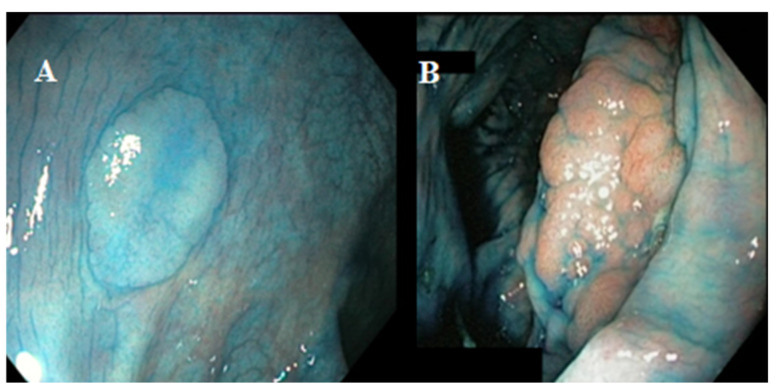
Endoscopic pictures of non-neoplastic (panel (**A**)) and neoplastic (panel (**B**)) lesions.

**Figure 4 cancers-14-04426-f004:**
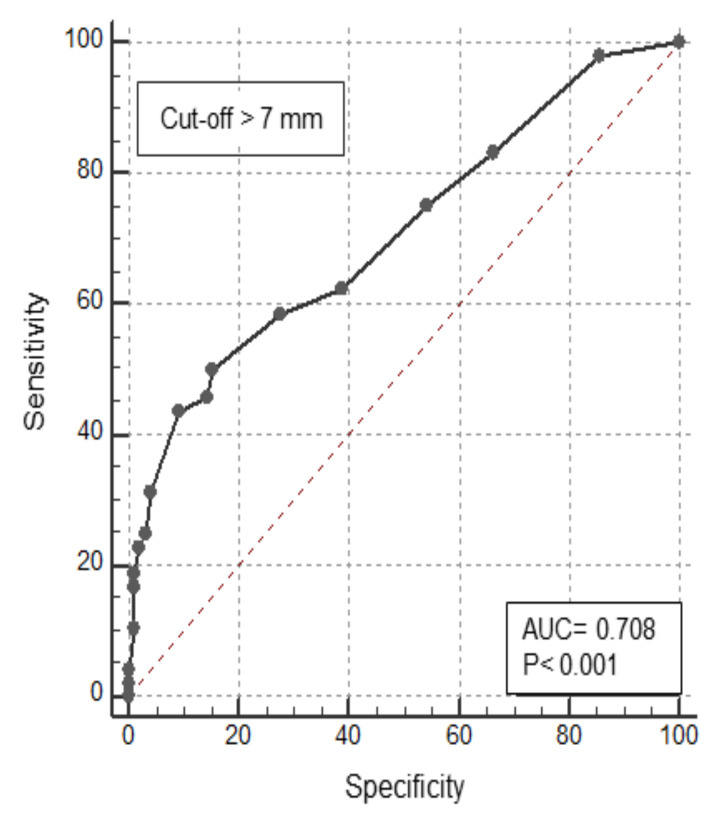
Receiver operating characteristic curve for dysplasia based on the lesion’s size.

**Table 1 cancers-14-04426-t001:** Demographic and clinical characteristics of 145 patients who underwent chromoendoscopy.

Characteristics of Patients	
Age (years); median (range)	53 (20–80)
Male, *n* (%)	82 (56.6)
Disease duration Median (range)	20 (2–62)
Age at disease onsetMedian (range)	30 (3–64)
UC, *n* (%)	125 (86.2)
*E2*	49 (39.2)
*E3*	76 (60.8)
CD, *n* (%)	16 (11.1)
*L2B*	9 (56.3)
*L2B2*	1 (6.2)
*L3B1*	5 (31.3)
*L3B2*	1 (6.2)
C-UNC, *n* (%)	4 (2.7)
Risk factors for CRC, *n* (%)	
-*High* (CSP, Family history of CRC, Dysplasia in the last 5 years)	30 (20.7)
*-Medium*	39 (26.9)
*-Low*	76 (52.4)
Previous therapy with steroids, ISS or biologics, *n* (%)	62 (42.8)
Ongoing therapy with steroids, ISS or biologics, *n* (%)	43 (29.7)
Smoking habit, *n* (%)	
*Yes*	18 (12.4)
*No*	90 (62)
*Former*	37 (25.6)
Familiar history of IBD, *n* (%)	
*Yes*	29 (20)
*No*	116 (80)

CRC = colorectal cancer; CSP = primary sclerosing cholangitis; ISS = Immunosuppressive drugs.

**Table 2 cancers-14-04426-t002:** Endoscopic characteristics of the patients undergoing DCE.

Disease Activity	
Mayo UC score, (*n*)	125
0, *n* (%)	72 (57.6)
1, *n* (%)	20 (16)
2, *n* (%)	22 (17.6)
3, *n* (%)	11 (8.8)
SES-CD, (*n*)	16
*Median, range*	1.5 (0–11)
Endoscopic remission in C-UNC, (*n*)	4
**BBPS ***	
*Median, range*	9 (6–9)
**Withdrawal Time (minutes)**	
*Median, range*	22 (8–70)
**Dye**	
Methylene Blue, *n* (%)	131 (90.3)
Indigo carmine, *n* (%)	14 (9.7)
**Colonoscopies with Visible Lesions;**	
*n* (%)	79 (54.5)

* BBPS = Boston Bowel Preparation Scale.

**Table 3 cancers-14-04426-t003:** Comparison between patients with dysplastic and non-dysplastic lesions.

	Patients with Dysplastic Lesions (*n* = 28)	Patients without Dysplastic Lesions (*n* = 117)	*p* Value
Age (years)			
Median (range)	62.5 (35–78)	53 (20–80)	0.0025
Male *n* (%)	17 (60.7)	65 (55.6)	0.67
Disease duration (years)			
Median (range)	19 (11–44)	20 (2–62)	0.59
Age at disease onset			
Median (range)	38 (5–62)	27.5 (3–64)	0.0012
IBD type			
UC	22 (78.6)	103 (88)	0.22
CD	5 (17.9)	11 (9.4)	0.19
C-UNC	1 (3.5)	3 (2.6)	0.58
UC extent, *n* (%)			
E2	7 (31.8)	42 (40.8)	0.48
E3	15 (68.2)	61 (59.2)	0.48
Risk factors for CRC, *n* (%)			
High (*CSP, Familiar history of CRC, Dysplasia in the last 5 years*)	13 (46.4)	17 (14.5)	0.0005
Medium	3 (10.7)	36 (30.8)	0.03
Low	12 (42.9)	64 (54.7)	0.29
Previous therapy with ISS or biologics, *n* (%)	8 (28.6)	54 (46.2)	0.13
Ongoing ISS therapy, *n* (%)	6 (21.4)	37 (31.6)	0.36
Smoker, *n* (%)			
Current or former	13 (46.4)	42 (35.9)	0.39
Never	15 (53.6)	75 (64.1)	0.39
Familiar history of IBD, *n* (%)	1 (3.5)	27 (23)	0.01

CRC = colorectal cancer; CSP = primary sclerosing cholangitis; ISS = immunesuppressive drugs.

**Table 4 cancers-14-04426-t004:** Predictive endoscopic features associated with risk of dysplasia.

Risk Factors	Univariate Analysis		Multivariate Analysis	
	OR (95% CI)	*p* Value	OR (95% CI)	*p* Value
Pit Pattern (Kudo Classification)				
-No neoplastic lesions (Kudo I-II)	0.92 (0.46 to 1.8)	0.82		-
-Neoplastic lesions (Kudo III-IV)	6.08 (2.77 to 13.4)	**<0.0001**	5.8 (2.3 to 14.6)	**0.0002**
Lesion’s size	1.13 (1.14 to 1.21)	**0.0002**	1.16 (1.06 to 1.3)	**0.0009**
Morphology (Parigi Classification)				
-Polypoid lesions (0-Is, 0-Isp, 0-Ip)	2.93 (1.39 to 6.21)	**0.0037**	7.4 (2.7 to 20.2)	**0.0001**
-No-polypoid lesions (0-IIa, 0-IIb-0-IIc, LST)	0.34 (0.16 to 0.72)	**0.0047**	0.13 (0.05 to 0.4)	**0.0001**
Lesion location				
-Cecum/right colon	1.42 (0.68 to 2.94)	0.35		-
-Transverse colon	1.07 (0.46 to 2.47)	0.86		-
-Left colon/sigmoid colon	0.68 (0.25 to 1.81)	0.44		-
-Rectum	0.11 (0.01 to 0.92)	0.04	0.19 (0.02 to 1.8)	0.15
Presence of mucosa inflammation around lesion	4.55 (0.39 to 52.2)	0.21		-
Lesion removal vs lesion biopsied	1.29 (0.58 to 2.88)	0.53		-

**Table 5 cancers-14-04426-t005:** Sensitivity, specificity, PPV, and NPV of single and combined endoscopic features to predict dysplasia.

	Sensitivity	Specificity	PPV	NPV
**Kudo III-IV**	79%	80%	58%	92%
**Lesion size > 7 mm**	46%	79%	53%	75%
**Polypoid Morphology**	67%	61%	41%	81%
**Kudo III-IV + lesion size > 7 mm**	67%	90%	87%	72%
**Kudo III-IV + Polypoid morphology**	87%	85%	75%	91%
**Polypoid morphology and lesion size > 7 mm**	71%	95%	83%	91%
**Kudo III-IV + polypoid lesions + lesion’s size > 7 mm**	90%	100%	100%	96%

PPV: positive predictive value. NPV: negative predictive value.

## Data Availability

The data that support the findings of this study are available from the corresponding author (G.M.), upon reasonable request.

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
