# Peer review of "Endoscopic Predictors of Neoplastic Lesions in Inflammatory Bowel Diseases Patients Undergoing Chromoendoscopy"

_cancers, 2022, doi:10.3390/cancers14184426_

Round 1

Reviewer 1 Report

In this manuscript, Lolli and colleagues aimed to identify predictive factors for dysplastic/neoplastic lesions in IBD patients. The authors recruited 219 patients and analyzed the dye based chromoendoscopy results. They have observed that kudo-pit pattern III-IV is a good predictor of dysplasia. Further, they have inferred that lesion size more than 7mm and lesions having all three endoscopic features are frequently dysplastic. All this information will be useful for clinicians to predict CRC based on dye-based chromo endoscopy results. However, I have the following comments.

         I.            The authors should explain the clinical terms such as Kudo pit pattern, and Paris classification briefly, which will be helpful for a reader not familiar with these endoscopic terms.

       II.            In addition, if the authors would provide some representative picture of predictive dysplastic vs non-dysplastic colons, it will be easier to follow.

     III.            The present form of the manuscript is unable to convince, how this manuscript is different from references 19 and 20. The authors should discuss how their findings will advance the knowledge.

    IV.            The authors should present graphs along with tables to depict their results.

Author Response

Reviewer 1 
We would like to thank the reviewer for his/her helpful comments and suggestions. In response to 
specific comments raised by this reviewer: 
1. The authors should explain the clinical terms such as Kudo pit pattern, and Paris classification briefly, which will be helpful for a reader not familiar with these endoscopic
Response: We thank the reviewer for his/her suggestion. The Kudo pit pattern and Paris Classification were added and described in materials and methods. 
2. In addition, if the authors would provide some representative picture of predictive dysplastic vs non-dysplastic colons, it will be easier to follow.
Response: As suggested, the endoscopic pictures representative of dysplastic vs nondysplastic lesions were added in the manuscript. 
3. The present form of the manuscript is unable to convince, how this manuscript is different from references 19 and 20. The authors should discuss how their findings will advance the knowledge.
Response: We thank the reviewer for his/her comments. The DCE role in endoscopy for dysplasia detection is now well known. For this reason, the aim of our study, unlike the mentioned studies, is focused on assessing different endoscopic parameters combining them to increase the dysplasia prediction in IBD patients undergoing surveillance DCE. 
In addition, in our study DCE was not compared with other techniques (virtual 
chromoendoscopy / high definition endoscopy). As suggested, in the discussion we have more widely discussed about this data. 
4. The authors should present graphs along with tables to depict their results. 
Response: We thank the reviewer for his/her suggestion but we avoided to included other graphs given that many tables and pictures are already in the manuscript. 

Reviewer 2 Report

This study was a prospective research to assess several endoscopic factors to effectively predict dysplastic/neoplastic lesions in IBD patients undergoing dye-based chromoendoscopy (DCE). The authors found that Kudo pit pattern III-IV, size >7 mm, and polypoid lesions were associated with a risk of dysplasia during surveillance colonoscopy in IBD patients.

Major comments

However, many studies have already demonstrated the effectiveness of DCE in IBD surveillance. In addition, these specific endoscopic findings during DCE have no clinical significance if they do not change the processes like decision of treatment or methods of endoscopic treatment that occur during surveillance endoscopy. Since these factors were unique shapes that can be observed during endoscopy, it is appropriate to submit the results of this study to an endoscopy-related journal.

Author Response

We would like to thank the reviewer for his/her evaluation. This reviewer pointed correctly that many studies have already demonstrated the effectiveness of DCE in IBD surveillance. We do agree with him/her and, indeed, we have widely indicated in the original and revised manuscript that the aim of our study was not to confirm the effectiveness of DCE but to identify predictors of dysplastic lesions. The reviewer also stated that these specific endoscopic findings during DCE have no clinical significance if they do not change the processes like decision of therapy or methods of endoscopic treatment that occur during surveillance endoscopy. We do not agree with the reviewer as identification of lesions which have high probability to be neoplastic can help endoscopists during the sampling process. Clearly, we are aware that our findings need further confirmation in larger and multicenter studies

Round 2

Reviewer 1 Report

No further comments.

Author Response

Assistant Editor: Rhea Feng

Cancers

Manuscript title:Endoscopic predictors of neoplastic lesions in inflammatory bowel diseases

patients undergoing chromoendoscopy”

Rome, 30.08.2022

Dear Rhea,

thank you for your letter dated 26 August 2022 and for giving us again the opportunity to improve and re-submit our manuscript. We would also like to thank the reviewers for their evaluation and helpful comments.

As indicated in the point-by-point reply to the reviewers’ comments below we have revised and expanded the manuscript taking into account all the issues raised by the reviewers. Since we hope these changes successfully address the reviewers’ comments, we would like to resubmit the paper for publication.

Changes are underlined in the revised version of the manuscript.

Yours Sincerely,

Giovanni Monteleone (on behalf of all the authors)

Department of Systems Medicine

University of Rome Tor Vergata, Via Montpellier, 1, 00133 Rome, Italy

Phone +39.06.72596158

Fax     +39.06.72596158

Email: gi.monteleone@med.uniorma2.it

Referee 1:

We would like to thank the reviewer for his/her positive evaluation.

Referee 2:

We would like to thank the reviewer for his/her helpful comments and suggestions. In response to specific comments raised by this reviewer:

  1. I agree with the authors as identification of lesions which have high probability to be neoplastic can help endoscopists during the sampling process. Please give explanation on what is the difference between high probability of dysplasia and low probability of dysplasia of lesion in sampling, and please explain how the strategy of treatment or observation is different.

Response: The endoscopic lesions that are dark or uneven redness and exhibit a nodular or villous pattern with  a depressed/ulcerated morphology and irregular vascular pattern are most likely dysplastic. Nevertheless, IBD related inflammation can modify the mucosa appearance thereby making difficult the detection and characterization of the colonic lesions. The presence of dysplastic lesions requires their complete removal and for patients with such lesions the intervals of endoscopic surveillance have to be closer.

  1. What I meant to say is that the endoscopic predictors the authors found are not novel. The Kudo pit pattern is widely used as a method to indirectly know the grade of histology. It is a well-known fact that the size of the tumor generally increases the neoplastic probability in the colon. It is a good research result to prepare new criteria on dysplastic lesions with increased sensitivity and specificity by combining existing known facts by the authors. However, since the authors did not discover any novel endoscopic features, it should be dealt with in the discussion section.
  2. Response: We are aware that some of the endoscopic factors described in our study had been previously associated with enhanced risk of dysplasia. However, our data indicate that combining more variables enhances the sensitivity and specificity to predict dysplasia. We discuss this point in the limits of study.

Reviewer 2 Report

Review 1st round

This study was a prospective research to assess several endoscopic factors to effectively predict dysplastic/neoplastic lesions in IBD patients undergoing dye-based chromoendoscopy (DCE). The authors found that Kudo pit pattern III-IV, size >7 mm, and polypoid lesions were associated with a risk of dysplasia during surveillance colonoscopy in IBD patients.

Major comments

However, many studies have already demonstrated the effectiveness of DCE in IBD surveillance. In addition, these specific endoscopic findings during DCE have no clinical significance if they do not change the processes like decision of treatment or methods of endoscopic treatment that occur during surveillance endoscopy. Since these factors were unique shapes that can be observed during endoscopy, it is appropriate to submit the results of this study to an endoscopy-related journal.

Minor comments

If subjects had had IBD for at least 8 years, they would have had a colonoscopy before that. What is the average duration of colonoscopy for this study after the first colonoscopy?

------------------------------------------------------------------------------------------------------------

Response to Review #1

We would like to thank the reviewer for his/her evaluation. This reviewer pointed correctly that many studies have already demonstrated the effectiveness of DCE in IBD surveillance. We do agree with him/her and, indeed, we have widely indicated in the original and revised manuscript that the aim of our study was not to confirm the effectiveness of DCE but to identify predictors of dysplastic lesions. The reviewer also stated that these specific endoscopic findings during DCE have no clinical significance if they do not change the processes like decision of therapy or methods of endoscopic treatment that occur during surveillance endoscopy. We do not agree with the reviewer as identification of lesions which have high probability to be neoplastic can help endoscopists during the sampling process. Clearly, we are aware that our findings need further confirmation in larger and multicenter studies

--------------------------------------------------------------------------------------------------------------

Review 2nd round

1. I agree with the authors as identification of lesions which have high probability to be neoplastic can help endoscopists during the sampling process. Please give explanation on what is the difference between high probability of dysplasia and low probability of dysplasia of lesion in sampling, and please explain how the strategy of treatment or observation is different.

2. What I meant to say is that the endoscopic predictors the authors found are not novel. The Kudo pit pattern is widely used as a method to indirectly know the grade of histology. It is a well-known fact that the size of the tumor generally increases the neoplastic probability in the colon. It is a good research result to prepare new criteria on dysplastic lesions with increased sensitivity and specificity by combining existing known facts by the authors. However, since the authors did not discover any novel endoscopic features, it should be dealt with in the discussion section.

Author Response

(The authors gave the same response as above.)

Round 3

Reviewer 2 Report

I think the authors have revised it correctly.